# Uncertainty-Aware Contour Proposal Networks for Cell Segmentation in Multi-Modality High-Resolution Microscopy Images

**Eric Upschulte**[1,2], **Stefan Harmeling**[3], **Katrin Amunts**[1,4], **Timo Dickscheid**[1,2,5]

[1] Institute of Neuroscience and Medicine (INM-1), Research Centre Jülich, Germany
[2] Helmholtz AI, Research Centre Jülich, Germany
[3] Department of Computer Science, Technical University Dortmund, Germany
[4] Cécile & Oskar Vogt Institute of Brain Research, University Hospital Düsseldorf, Germany
[5] Department of Computer Science, Heinrich Heine University Düsseldorf, Germany

## Abstract

We present a simple framework for cell segmentation, based on uncertainty-aware **Contour Proposal Networks** (CPNs). It is designed to provide high segmentation accuracy while remaining computationally efficient, which makes it an ideal solution for high throughput microscopy applications. Each predicted cell is provided with four uncertainty estimations that give information about the localization accuracy of the detected cell boundaries. Such additional insights are valuable for downstream single-cell analysis in microscopy image-based biology and biomedical research. In the context of the NeurIPS 22 Cell Segmentation Challenge, the proposed solution is shown to generalize well in a multi-modality setting, while respecting domain-specific requirements such as focusing on specific cell types. Without an ensemble or test-time augmentation the method achieves an F1 score of $0.8986$ on the challenge's validation set. Code is available at `https://github.com/FZJ-INM1-BDA/neurips22-cell-seg`.

## 1 Introduction

Object detection and segmentation in images is fundamental in many research areas. Solutions need to be both reliable and efficient, as downstream tasks can be sensitive to segmentation and detection quality and often involve large quantities of data that need to be processed. This work is based on the Contour Proposal Network (CPN) [1], which models instance segmentation as a *sparse detection* problem by performing regression of object contours anchored at pixel locations. This way, the model is capable of assigning multiple objects to the same pixel and thus recover partially superimposed objects with their actual shape, which is highly favorable for shape-sensitive downstream tasks like morphological cell analysis. The CPN uses a backbone network to extract multi-scale feature maps from the input image, from which regression heads generate candidate contour representations at each pixel, while a classification head determines whether an object is present or not at these locations. Based on the classifications, a proposal sampling stage then extracts a sparse list of contour representations, which are explicitly projected to the pixel domain using the differentiable Fourier transformation and thus learn to encode contour representations in the frequency domain. Precision of contours is further optimized by applying a displacement field in the pixel domain which is the output of an additional regression head. As an extension to the original CPN, we add dedicated supervision of boundaries and propose an additional branch which estimates localization uncertainty for boundaries. The latter is a technique originally intended for anchor-free object detection with bounding boxes [2]. The complete framework is trained end-to-end, and complemented by a final uncertainty-aware non-maximum suppression (NMS) [2] to remove redundant detections.

36th Conference on Neural Information Processing Systems (NeurIPS 2022).

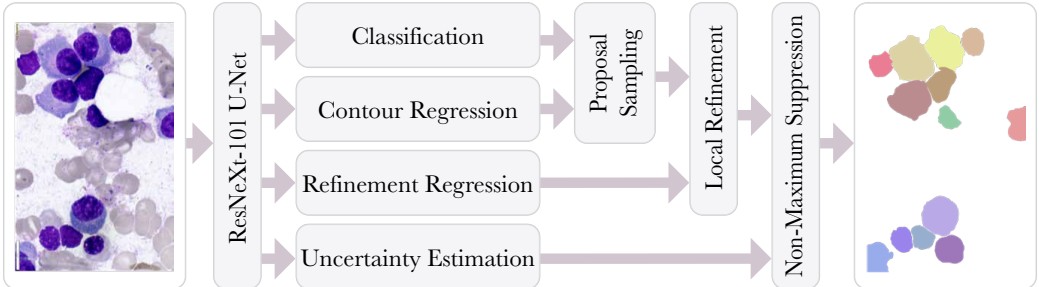

Figure 1: Overview of the uncertainty-aware Contour Proposal Network. The CPN [1] uses a U-Net [3] architecture with a ResNeXt-101 [4] encoder to compute multi-scale features based on the input image. Using these features, a classification head determines if an object of interest is present in the input image at a given pixel location. The contour regression head encodes complete and closed object contours sparsely. A list of contour proposals can be extracted using the classification results. A local refinement step [1] is applied to improve pixel precision of the contours. The uncertainty head estimates four localization uncertainties for proposed object boundaries. This information can be used in downstream tasks and is combined with the classification score for the final uncertainty-aware NMS.

## 2 Methods

### 2.1 Preprocessing

All image data is first converted to 8-bit unsigned integer (`uint8`) arrays. For this we explore two components: (a) A percentile-based min-max normalization, and (b) a mean-intensity based gamma correction. The latter is intended to improve contrast especially in dark images, by applying stronger corrections for smaller mean-intensities. Different conversion strategies are applied during training, as a form of data augmentation. Some datasets contain fractured annotations. As the proposed method assumes closed object contours, all objects that are annotated with more than one connected component are flagged and ignored during training. Also, large objects are automatically removed from the Cellpose dataset [5], to prevent negative influence of irregular shapes.

### 2.2 Contour Proposal Networks

For the detection and segmentation of cells we use CPNs [1]. The original CPN uses a backbone network to produce features based on the input image for multiple prediction heads: A classification head that decides whether a pixel represents an object, a contour regression head that predicts object contour coordinates in vector format, a location regression head that estimates the offset between a pixel position and actual object location, and a refinement head that fine-tunes pixel accuracy of the contours from a learned 2D deformation field. In this work, we make a specific design choice for the backbone network, and propose an additional uncertainty estimation head that predicts the uncertainty with respect to localization accuracy of object boundaries.

### 2.2.1 Classification

The classification head classifies each pixel into $K + 1$ categories, namely background and $K$ foreground object classes. Note that this is technically over-parameterized and can be reduced to a $K$-class classification problem [6]. In this work we focus on just one object class, hence the classification head is a binary classifier. As the CPN relies on sparse classification, only a centric fraction of an object's area is marked with a respective foreground label. As a loss function the weighted Binary Cross Entropy (BCE) is used:

$$\mathcal{L}_{\text{object}} = -w_{\text{fg}} o \log \hat{o} - w_{\text{bg}} (1 - o) \log(1 - \hat{o}) \tag{1}$$

with $o$ and $\hat{o}$ denoting the targeted and predicted class score, respectively. For legibility we specify all losses per pixel, and imply averaging. To compensate for the class imbalance between object and background pixels, losses of each set of class pixels are reduced separately.

### 2.2.2 Contour prediction

Much like bounding boxes, contours can be represented in a vector format [1]. The contour regression head predicts entire object contours as a vector embedding at each pixel location. During inference, the classification head decides if a contour vector represents an object in the input image. In [1] the authors follow [7] and define contours as a series of 2d coordinates $((x_1, y_1), \ldots, (x_S, y_S))$ using the Fourier sine and cosine transformation

$$
\begin{aligned}
x_s &= a_0 + \sum_{n=1}^{N} \left( a_n \sin\left(\frac{2n\pi t_s}{T}\right) + b_n \cos\left(\frac{2n\pi t_s}{T}\right) \right) \\
y_s &= c_0 + \sum_{n=1}^{N} \left( c_n \sin\left(\frac{2n\pi t_s}{T}\right) + d_n \cos\left(\frac{2n\pi t_s}{T}\right) \right)
\end{aligned}
\tag{2}
$$

with $N$ denoting the order hyperparameter, $a_n, b_n, c_n$ and $d_n$ denoting contour coefficients that are predicted by the network, and the location parameter $t_i \in [0, 1]$ with interval length $T = 1$ that defines the location on the contour for which coordinates are to be calculated. An entire contour can be sampled by examining Eq. 2 at $t_1, \ldots, t_S$ with $t_i < t_{i+1}$. The vector size $4N + 2$ of the embedding is controlled by the order $N$. By definition of Eq. 2, it also regulates how precise the approximation of a contour is. Larger settings of $N$ yield larger embeddings and allow for more details to be preserved, while smaller settings focus on basic shape characteristics. As in [1], the contour regression loss is defined as the average absolute difference between predicted and targeted contour coordinates:

$$
\mathcal{L}_{\text{contour}} = \frac{1}{2S} \sum_{s=1}^{S} (|x_s - \hat{x}_s| + |y_s - \hat{y}_s|).
\tag{3}
$$

### 2.2.3 Boundary supervision

As a further incentive for correct localization, object boundaries are supervised using the generalized IoU (GIoU) loss [8] denoted by $\mathcal{L}_{\text{boundary}}$. For this purpose, boundaries are derived from contours in the form of bounding boxes. As in [1], a bounding box is defined as $(\min_s x_s, \min_s y_s, \max_s x_s, \max_s y_s)$.

### 2.2.4 Local refinement

In [1] local refinement was proposed to maximize the pixel precision of the predicted contours. Effectively, local refinement allows the network to self-correct by leveraging local features that describe parts of an object, to improve a contour that describes the outlines of a whole object. Assuming that local refinement improves contour precision, this step also provides self-supervision potential for the contour regression, relevant for semi-supervised scenarios. As in [1], the refinement loss $\mathcal{L}_{\text{refine}}$ is the absolute L1 distance between refined and target contours.

### 2.2.5 Localization uncertainty estimation

An additional branch is added to the CPN architecture to estimate the uncertainty of object boundaries. Following [2, 9], four uncertainties (top, right, bottom, left) are predicted per object. The uncertainty branch is trained using the negative power log-likelihood loss (NPLL) [2]

$$
\mathcal{L}_{\text{uncertainty}} = \eta \left( \sum_i \left( \frac{(v_i - \hat{v}_i)^2}{2\delta_i^2} + \frac{1}{2} \log \delta_i^2 \right) + 2 \log 2\pi \right)
\tag{4}
$$

with $i$ denoting the four boundaries (top, right, bottom, left), $v_i$ and $\hat{v}_i$ denoting the true and predicted boundaries, $\delta_i$ denoting the estimated uncertainty and $\eta$ the IoU score between predicted and ground-truth box. In practice, $\delta_i$ is sigmoid activated and scaled by a constant. During inference the scaling is omitted, such that the estimated uncertainty lies within the interval $[0, 1]$.

### 2.2.6 Training objective

The complete per-pixel objective minimized during training is given by

$$
\mathcal{L} = \mathcal{L}_{\text{object}}(o, \hat{o}) + o(\lambda_0 \mathcal{L}_{\text{uncertainty}} + \lambda_1 \mathcal{L}_{\text{contour}} + \lambda_2 \mathcal{L}_{\text{refine}} + \lambda_3 \mathcal{L}_{\text{repr}} + \lambda_4 \mathcal{L}_{\text{boundary}})
\tag{5}
$$

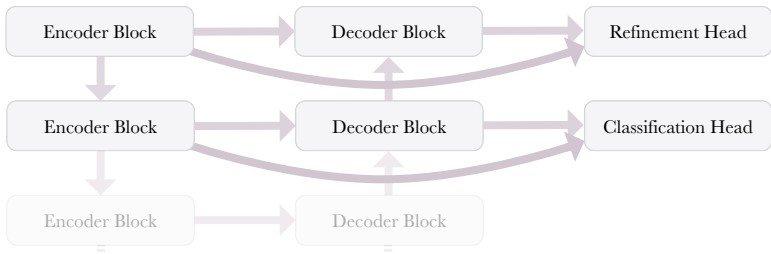

Figure 2: Lateral skip connection between encoder blocks, decoder blocks and classification and regression heads, respectively.

with $o$ and $\hat{o}$ denoting targeted and predicted binary classification scores, $\lambda_i$ loss-specific weights and $\mathcal{L}_{\text{repr}}$ denoting an additional regularization that minimizes the distance between targeted and predicted contour embeddings in the frequency domain [1].

### 2.2.7 Uncertainty-aware non-maximum suppression

The CPN applies NMS to remove redundant contour proposals. Vanilla NMS specifically keeps proposals with a high classification score and suppresses proposals with lower scores and Intersection over Unions (IoUs) larger than a given threshold. However, there is no guarantee that the classification score alone is an appropriate indicator for the quality of a proposal. We adopt the strategy proposed in [2] and use the product of score and certainty $\hat{o}(1 - 1/4(\sum_i \delta_i))$ to include uncertainty estimations into the selection process of NMS instead of choosing proposals based on the classification score $\hat{o}$.

### 2.2.8 Backbone

The backbone architecture of the CPN is a U-Net [3] with a ResNeXt-101 [4] encoder. The final encoder features are passed to a pyramid pooling module (PPM) [10] that drastically increases field-of-view and provides global context. The decoder consists of residual blocks [11] with projection shortcuts [11], group normalization [12] and Leaky-ReLU activation [13]. To mitigate the development of artifacts stemming from other heads and to leverage shallow features, we add an additional lateral skip-connection between encoder and the refinement and classification head, respectively. Figure 2 illustrates the setup. Especially for the classification and refinement tasks, we view shallow features as vital, as these tasks often depend on local edge and color information. The architecture is initialized randomly and trained from scratch.

### 2.2.9 Semi-supervised boundaries with Listen2Student

To include unlabeled examples in the training, we use the uncertainty-aware Listen2Student (L2S) mechanism [9]. A teacher model produces bounding boxes as pseudo-labels, which are used to supervise the student model. In Listen2Student, the student only learns from the teacher if $\delta_i^s > \sigma^s$ and $\delta_i^t \leq \delta_i^s - \sigma^m$, with $i$ denoting the boundary index as in Eq. 4, $\delta_i^s$ and $\delta_i^t$ denote the uncertainties of student and teacher, and $\sigma^s$, $\sigma^m$ denote the uncertainty threshold and margin hyperparameters, respectively. If this condition is met, the absolute difference between student and teacher boundary is minimized.

## 2.3 Efficiency

To improve model inference speed and reduce resource consumption the solution utilizes automatic mixed precision (AMP) via PyTorch's *autocast* feature. It automatically selects operation-specific data types to improve performance while aiming to maintain accuracy[1]. Additionally, we leverage that contours can be defined on an arbitrary resolution without losing precision and apply the CPN heads with features of lower resolutions. This decreases computational costs and thus further improves inferences speed.

---

[1] `https://pytorch.org/docs/1.12/amp.html`

## 2.4 Post-processing

The detected object contours are converted to label images using rasterization and region filling. Since contours may overlap and the applied datasets do not allow overlap, only unambiguous regions are filled initially. We chose to fill overlapping regions using region growing [14] seeded by the initial labels.

# 3 Experiments

## 3.1 Datasets

The following datasets were used for training : BBBC039 [15], BBBC038 [16], Omnipose [17], Cellpose [5], Sartorius Cell Instance Segmentation (SCIS) [2], Livecell [18], NeurIPS 22 - Cell Segmentation in Multi-modality Microscopy Images [3].

## 3.2 Implementation details

### 3.2.1 Environment settings

The development and validation environments are presented in Table 1. Note that the development environment lists specifications of a single node of the JURECA HPC system [19]. Multiple nodes were used during development. Validation is performed using a dockerized inference script. Following the challenge guidelines[4], the Docker environment is limited to 28 GB of RAM and Docker's default shared memory size of 64 MB[5]. As minimum requirement for inference, we recommend single image processing with a patch size of $512 \times 512$, which yields peak GPU memory consumption of 2.25GiB[6] (3.51GiB on process level) and 20.23 FPS with AMP enabled in the validation environment.

Table 1: Development and validation environment.

|  | Development | Validation |
|---|---|---|
| System | Rocky Linux 8 | Ubuntu 20.04 |
| CPU | 2× AMD EPYC 7742 | Intel® Core™ i7-7800X |
|  | $2 \times 64$ cores, 2.25 GHz | $12 \times 3.50$ GHz |
| RAM | $16 \times 32$ GB, 3200 MHz | $8 \times 16$ GB |
| GPU (number and type) | 4× NVIDIA A100 | 1× NVIDIA RTX 2080 Ti |
|  | $4 \times 40$ GB HBM2e |  |
| CUDA version | 11.3 | 11.3 |
| Programming language | Python 3.8.12 | Python 3.8.12 |
| Deep learning framework | PyTorch 1.12 | PyTorch 1.12 |
| Specific dependencies | celldetection [7] | Docker [8] |

### 3.2.2 Training protocols

**Training schedule**     Inspired by evolutionary optimization strategies and stochastic gradient descent with warm restarts (SGDR) [20], we train multiple models with different hyperparameters, select the best performing models and restart training with changed hyperparameters. This process is repeated until validation scores no longer improve. Changed hyperparameters specifically regard learning rate, augmentation policy, the use of random data subsampling, as well as the size of the centric fraction of an object area that is trained to represent the object. In the progression of restarts, the latter is increased, learning rate decreased and subsampling disabled. Hyperparameters that regard the architecture are not altered during this process. The used architectures are initialized randomly and

---

[2]https://www.kaggle.com/competitions/sartorius-cell-instance-segmentation/data

[3]https://neurips22-cellseg.grand-challenge.org

[4]https://neurips22-cellseg.grand-challenge.org/metrics/

[5]https://docs.docker.com/engine/reference/run

[6]https://pytorch.org/docs/stable/generated/torch.cuda.max_memory_allocated.html

[7]https://celldetection.org

[8]https://docker.com

trained from scratch. All listed datasets were used jointly during training. Data from the NeurIPS 22 - Cell Segmentation challenge (NeurIPS22 CellSeg Challenge) was naively oversampled by duplicating each example five times to increase the frequency with which it occurs in training batches. As a naive form of randomized data pruning, we applied random subsampling of the joint dataset during the first two training generations. Consequently, random examples can be over- or undersampled. During training, the data is retrieved in random order. Training protocol details are listed in Table 2. Following [21], we scale the learning rate by $\sqrt{bs/64}$, with bs denoting the total batch size. Except the contour loss weight $\lambda_1 = 1.5$, all factors $\lambda_i$ are set to 1.

**Data augmentation** Data augmentation is performed online, according to a predefined policy. Grayscale and RGB images have specific augmentation pipelines, respectively. Augmentations include rotation, rescaling, flipping, blurring, gamma transformation, noise, HSV shifting, color mapping and channel shuffling. For the rescaling augmentation, an instance-aware, dynamic approach was used. Annotated object sizes are examined on-the-fly to define rescale bounds for the otherwise predominantly drastic rescale operation. All images are randomly cropped to $512 \times 512$ after all augmentations were performed.

### 3.2.3 Inference protocols

For inference, a single model is selected. It is applied with AMP via PyTorch's *autocast* feature, which automatically selects operation-specific data types to improve performance. Large images are processed using a sliding window approach with a window size of $768 \times 768$ and a stride of $384 \times 384$. Redundant detections, stemming from the overlap of the sliding windows, are removed using uncertainty-aware NMS (see Section 2.2.7), just as it is done inside the model. To remain efficient, the overlap is the only explicitly performed redundant prediction. Hence, no further test-time augmentation (TTA) is applied.

Table 2: Training protocols.

| | |
|---|---|
| Network initialization | "he" normal |
| Batch size (bs) | 10/GPU |
| Patch size | 512×512×3 |
| Total epochs | 225 |
| Optimizer | Adam ($\beta_1 = 0.9$, $\beta_2 = 0.999$, $\lambda = 0.00002$) |
| Initial learning rate (lr) | lr $\sim \mathcal{N}(0.0008, 0.0001)$ |
| Lr decay schedule | MultiStepLR (1000 warmup steps, $\gamma = 0.666$) |
| Training time | 17.08 hours |
| Loss function | BCE, L1, NPLL, GIoU |
| Number of model parameters | 228.24 M |
| Number of flops | 443.60 GF[9] |

### 3.3 Quantitative evaluation of segmentation accuracy

To quantitatively evaluate the detection and segmentation performance of the proposed method, the harmonic mean of precision and recall $F1_\tau = TP_\tau/(TP_\tau + 1/2(FP_\tau + FN_\tau))$ is used. TP, FP, FN denote true positives, false positives and false negatives, respectively. The IoU threshold $\tau \in [0, 1]$ determines if a detected object is counted as a match when compared to an object from the ground truth annotation. Results are reported for the validation set, of which unlabeled images are publicly available, as well as the non-public test dataset of the NeurIPS22 CellSeg Challenge.

### 3.4 Quantitative evaluation of efficiency

To quantitatively evaluate the efficiency of the proposed solution, the running time of a dockerized inference script is examined[10]. The inference time measures the entire run time of a `docker run` command that applies the proposed solution to a single image. Notably, this includes a relatively large overhead for environment setup.

---

[9]Measured with specified patch size using fvcore (https://github.com/facebookresearch/fvcore)
[10]https://neurips22-cellseg.grand-challenge.org/metrics

Table 3: Quantitative results on validation set. The detection and segmentation quality is reported in terms of $F1_{\tau=0.5}$, with $\tau$ denoting the IoU threshold. The results were obtained from the evaluation server of the NeurIPS22 CellSeg Challenge. *UA* and *L2S* abbreviate *uncertainty-aware* and *Listen2Student*, respectively. Results were achieved with automatic mixed precision (AMP) and local refinement enabled.

| Model | Backbone | Epochs | UA | GIoU | $F1_{\tau=0.5}$ |
|---|---|---|---|---|---|
| CPN | ResNeXt-101 U-Net | 345 | ✓ | × | 0.8815 |
| CPN | ResNeXt-101 U-Net | 315 | × | ✓ | 0.8848 |
| CPN$_{L2S}$ | ResNeXt-101 U-Net | 300 | ✓ | ✓ | 0.8961 |
| CPN | ResNeXt-101 U-Net | 225 | ✓ | ✓ | 0.8981 |
| 2 CPNs | ResNeXt-101 U-Net | 225, 195 | ✓ | ✓ | 0.9004 |

Table 4: Quantitative results for multiple UA-CPNs on validation set. The detection and segmentation quality is reported for different configurations of uncertainty-aware CPNs in terms of $F1_{\tau=0.5}$, with $\tau$ denoting the IoU threshold. The results were obtained from the evaluation server of the NeurIPS22 CellSeg Challenge. *UA*, *AMP* and *Refine* abbreviate *uncertainty-aware*, *automatic mixed-precision* and *local refinement*, respectively. GIoU loss was used in all configurations.

| Model | Backbone | Epochs | UA-NMS | AMP | Refine | $F1_{\tau=0.5}$ |
|---|---|---|---|---|---|---|
| CPN | ResNeXt-101 U-Net | 160 | × | ✓ | ✓ | 0.8447 |
| CPN | ResNeXt-101 U-Net | 160 | ✓ | ✓ | ✓ | 0.8457 |
| CPN | ResNeXt-101 U-Net | 225 | ✓ | ✓ | × | 0.8809 |
| CPN | ResNeXt-101 U-Net | 225 | ✓ | ✓ | ✓ | 0.8981 |
| CPN | ResNeXt-101 U-Net | 225 | ✓ | × | ✓ | 0.8983 |
| CPN | ResNeXt-101 U-Net | 225 | × | ✓ | ✓ | 0.8986 |

## 4 Results and discussion

### 4.1 Quantitative results on validation set

Quantitative results examining the influence of uncertainty estimation, GIoU loss, the Listen2Student mechanism and an ensembles are reported for the validation set of the NeurIPS22 CellSeg Challenge in Table 3. Different configurations of uncertainty-aware CPNs are evaluated in Table 4. The largest improvement in the reported results is achieved by increasing the number of epochs used for training, as can be seen in Table 4. Increasing epochs from 160 to 255 improves the $F1_{\tau=0.5}$ score by 0.0524, with *UA*, *AMP* and *Refine* enabled. The use of AMP, in the reported case, decreases the $F1_{\tau=0.5}$ score by 0.0002. Uncertainty-aware NMS slightly improves the score for a lower epoch count by 0.001. However, for a higher epoch count it decreases the score by 0.0005. Enabling local refinement improves the score by 0.0172. A simple ensemble of two CPNs achieves the highest score of 0.9004. In the conducted experiments the semi-supervised Listen2Student approach could not surpass fully-supervised training. Since neither Listen2Student, nor uncertainty-aware NMS are able to significantly improve the $F1_{\tau=0.5}$ score, we hypothesize that boundary localization is not the primary cause of errors for this setting of the IoU threshold $\tau$. Influences on scores with larger thresholds, as reported in [9], are not reflected in the evaluation of this challenge. The ablation of GIoU loss and uncertainty prediction shows that the joint use of both components increases F1 scores in this experiment. The results suggest that boundary localization uncertainty estimation and GIoU based boundary supervision are especially effective when used in a complementary manner. Overall, the results underline that longer training and local refinement [1] significantly improve $F1_{\tau=0.5}$ scores. Notably, local refinement is primarily intended to improve pixel accuracy, which predominantly improves $F1_{\tau}$ scores with greater IoU threshold $\tau$ [1].

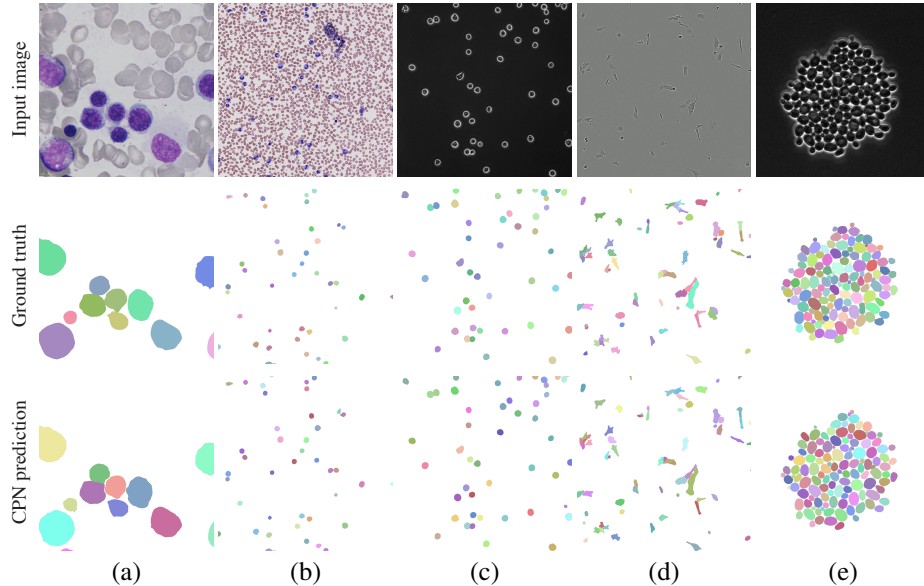

Figure 3: Segmentation examples. *(best viewed digitally)* Top row shows examples from a validation split of the NeurIPS22 CellSeg Challenge training set. Second row shows ground truth annotations and bottom row example segmentations from a CPN. Colors are chosen randomly per instance.

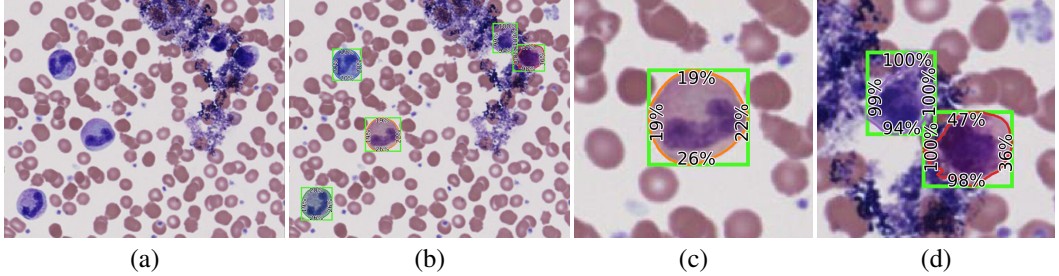

Figure 4: Examples with estimated localization uncertainties. *(best viewed digitally)* The examples above show crops of the image depicted in 3b. (b)-(d) add superimposed contours, bounding boxes and uncertainties predicted by a CPN. (c) and (d) show magnified examples for low and high localization uncertainties, respectively.

## 4.2   Qualitative results on validation set

To get a first impression of the performance, we considered a large sample of good and bad results qualitatively, such as the examples from the NeurIPS22 CellSeg Challenge validation set in Figure 5. Overall, the CPN seems to detect the correct objects with high accuracy across all domains of the validation set under various contrast and lighting conditions. The model is apparently able to capture domain-specific objectives during the training process. While some domains are more permissive and require the network to detect almost all object-like patterns (e.g. Figure 5d), others include a mixture of foreground and background objects, requiring the model to focus only on objects of interest and accept structures with high "objectness" in the background class (e.g. Figure 5b). Most of the erroneous predictions fall into one of three categories: (i) False positive; (ii) False negative; (iii) High boundary uncertainty. False positives typically occur if "object-like" patterns are present in the data. Figure 5c gives an example. High boundary uncertainty can cause contours to be partially imprecise. In extreme cases, this can cause overlap with adjacent objects, which in turn can produce false negatives during NMS. Notably, many of these error modes do not seem to be systematic, hence different instantiations of the same architecture do not necessarily struggle in the same situations. This indicates the potential for improvement using ensemble-like strategies.

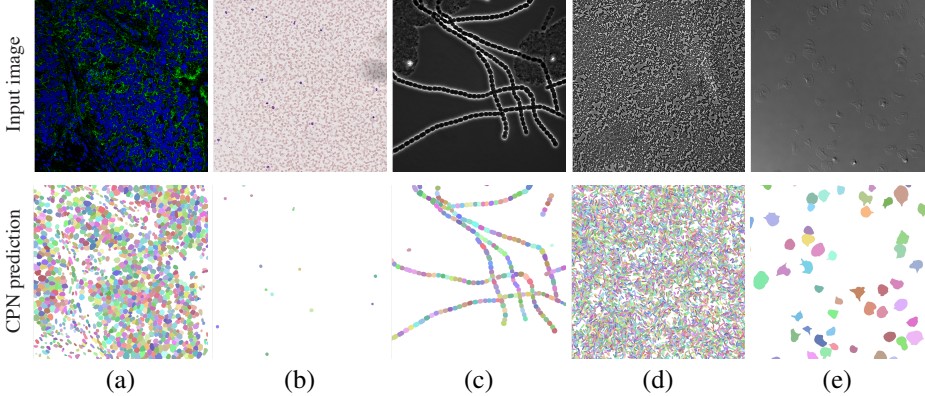

(a)         (b)         (c)         (d)         (e)

Figure 5: Segmentation examples from validation set. *(best viewed digitally)* Top row shows examples from the NeurIPS22 CellSeg Challenge validation set. Bottom row shows example segmentations from a CPN. Colors are chosen randomly per instance.

### 4.3 Segmentation efficiency results on validation set

To evaluate the efficiency of the proposed method we measure inference time using the protocol described by the NeurIPS22 CellSeg Challenge[11]. Details regarding the inference protocol and environment are given in Sections 3.2.3 and 3.2.1, respectively. Results for the validation set of the challenge are reported in Figure 6. All measured timings are below the time limits provided as part of the challenge. On average, the inference time is approximately $64\%$ of the given time limit. For the largest example with an equivalent size of 9398px it is only $19\%$ of the time limit. This underlines that the proposed solution is capable of handling high-throughput scenarios.

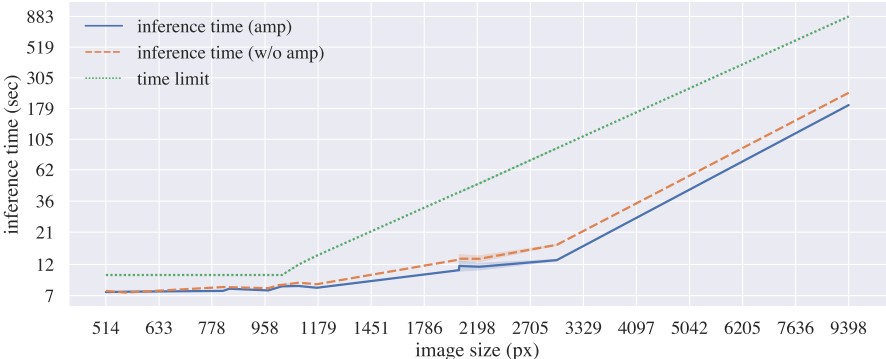

Figure 6: Inference times of the proposed solution with Docker. *(log scale)* Inference times in seconds are reported with equivalent image size $s := \sqrt{hw}$ in pixels, with $h$, $w$ denoting actual image dimensions. We contrast measures with AMP enabled *solid line* and disabled *dashed line*. The *dotted line* shows the time limit of the challenge. Note that inference times include Docker overhead.

### 4.4 Results on final testing set

The quantitative performance of a submitted model was evaluated by the organizers of the NeurIPS22 CellSeg Challenge on an undisclosed testing set that includes differential interference contrast (DIC), bright-field (BF), fluorescence (Fluo) and phase contrast (PC) imaging examples. The submitted model is a single uncertainty-aware CPN using uncertainty-aware NMS, AMP, local refinement and GIoU loss. The organizers provided results in the form of mean and median $F1_{\tau=0.5}$ scores, which are reported in Table 5. With a median F1 score of $0.8031$, the DIC imaging achieves the lowest

---

[11]https://neurips22-cellseg.grand-challenge.org/metrics/

Table 5: Quantitative results on undisclosed testing set. The detection and segmentation quality is reported in terms of $F1_{\tau=0.5}$, with $\tau$ denoting the IoU threshold. The results were provided by the challenge organizers. Modalities include differential interference contrast (DIC), bright-field (BF), fluorescence (Fluo) and phase contrast (PC) imaging.

| Reduction | F1-All | F1-BF | F1-DIC | F1-Fluo | F1-PC |
|-----------|--------|-------|--------|---------|-------|
| Median    | 0.8448 | 0.8372 | 0.8031 | 0.8136 | 0.9029 |
| Mean      | 0.8181 | 0.8253 | 0.7732 | 0.7922 | 0.8594 |

scores among the reported categories. Highest scores are achieved in the phase contrast imaging category with a median of 0.9029.

## 4.5 Limitation and future work

Based on the results and experiences during development, we infer several potential strategies for further improvement of segmentation performance. While we found during development that too extreme augmentation may harm the performance of the classification head of the proposed architecture, we hypothesize that it can benefit the contour proposal task. In particular, strong augmentations, such as the extensive use of color mapping, caused the network to occasionally struggle with excluding non-relevant objects, showing a tendency to provide contours for "object-like" background patterns for which it was not trained. While this is often a desired property to achieve good generalization, such behavior is inappropriate in some specific domains. Hence, we note as future work to examine whether the use of strongly augmented examples for the contour regression and mildly augmented examples for the entire model during training can improve contour regression generalization, while allowing the network to capture domain-specific behavior. On the production side, it may be relevant to apply trained models to similar but different domains. If it comes to generalizing to other domains, a fundamental question is which domain from the training set a new domain is the closest to, as the model may generalize learned domain-specific behavior, which may or may not be desired for new domains.

## 5 Conclusion

We proposed an uncertainty-aware Contour Proposal Network that detects and segments objects by proposing contours, equipped with boundary-specific uncertainty estimations. The proposed method can leverage the benefits of a model with relatively high parameter count, while remaining computationally efficient. The inference running times remain significantly below the time limits provided by the NeurIPS22 CellSeg Challenge, especially when applied to whole-slide images. Detection and segmentation performance is shown to be competitive, as our submission represents the second-best team according to the preliminary, public validation set leaderboard and the third best team according to the final testing set.

## Acknowledgement

This project received funding from the European Union's Horizon 2020 Research and Innovation Programme, grant agreement 945539 (HBP SGA3), and Priority Program 2041 (SPP 2041) "Computational Connectomics" of the German Research Foundation (DFG). This work was also funded by Helmholtz Association's Initiative and Networking Fund through the Helmholtz International BigBrain Analytics and Learning Laboratory (HIBALL) under the Helmholtz International Lab grant agreement InterLabs-0015. The authors gratefully acknowledge the computing time granted through JARA on the supercomputer JURECA [19] at Forschungszentrum Jülich.

The authors of this paper declare that the segmentation method they implemented for participation in the NeurIPS 2022 Cell Segmentation challenge has not used any private datasets other than those provided by the organizers and the official external datasets and pretrained models. The proposed solution is fully automatic without any manual intervention.

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
