# OpenReview forum: "Uncertainty-Aware Contour Proposal Networks for Cell Segmentation in Multi-Modality High-Resolution Microscopy Images"
_NeurIPS.cc/2022/Challenge/CellSeg — Submitted to NeurIPS CellSeg 2022_

### Official Review · Reviewer_y7jn · 2022-12-28
**Well structured and written, only minor suggestions for improvement**

**Rating:** 7
**Confidence:** 5

**Review:**

## Overall
The authors propose Uncertainty-Aware Contour Proposal Networks (CPNs) for instance segmentation of cells. CPNs use an object detection-like approach by learning parameters of fourier shape descriptors to approximate the shapes of cells. The authors extend their previous work on CPNs by adding a learned uncertainty - quantification. The predicted uncertainties are then also used in their NMS post-processing pipeline to further improve segmentation performance.

The paper is well structured and written. It contains all information necessary to reproduce most of their results and also a link to their code base. Here are some suggestions to improve the paper

## More details in 3.2.2. Training schedule

- 3.2.2. ", we train multiple models with different hyperparameters, including augmentation strategies, select the best performing models and restart training with changed hyperparameters": This could be made more transparent to increase reproducibility. Which hyperparameters (besides augmentation strategies) were changed. If you have the F1 scores at hand for a model trained without the SGDR scheme, you could add it to the ablation study in Table 3.

- 3.2.2. "Data from the NeurIPS 22 - Cell Segmentation challenge was oversampled": the exact sampling protocol could be described here

## Ablation study in Table 3 could be more organized
- All models in the table are uncertainty-aware models, thus the "UA" column has checkmarks everywhere. So this column is irrelevant and can be deleted.

- There are three messages mixed up in Table 3:
1. More training epochs lead to models that show better performance
2. Model ensembling further improves upon the best models
3. Ablation: your proposed changes indeed increased performance

The first two messages could be communicated in a single table, while the ablation study should be put in a separate table. In addition the ablation study would be more informative if all models have the same number of training epochs, which could be done without retraining (e.g. just show the 225 epoch checkpoint, there only the UA-NMS ablation is missing).

---

### Official Review · Reviewer_LNiA · 2023-01-05
**Well constructed, impressive performance.**

**Rating:** 8
**Confidence:** 4

**Review:**

The authors proposed Uncertainty-aware CPNs for instance segmentation of cells. This work is based on their previous Contour Proposal Network and the uncertainty-aware mechanism is introduce to improve the performance. I am impressed by its strong capability on distinguishing overlapped cells and its generalization ability in such multi-modal scenes. In general, this paper is well written and contains sufficient details for reproduction.

However, there are still some minor suggestions. In ablation study, I hope to see the effectiveness of UA module since it is added in this work. What's more, if more visualization during the training process like the output of the four heads in the model can be shown , it would be better for readers.

---

### Official Review · Reviewer_dj2B · 2023-01-07
**Clean and unified solution as well as strong results**

**Rating:** 9
**Confidence:** 5

**Review:**

Summary:
The authors present a novel framework for multi-modal cell segmentation. The fundamental segmentation structure is a contour proposal network. Besides, the authors propose an uncertainty estimation and new contour supervision to improve the segmentation accuracy. Finally, the framework achieves great segmentation performance and efficiency.

Pros:
1. Deep and detailed introduction of the implemented methods.
2. Good extension to the original CPN by adding new boundaries supervision and localization uncertainty estimation.
3. Show the limitation of a semi-supervised method using the unlabeled images.
4. Qualitative and quantitative results show the effectiveness and efficiency of the proposed framework.
5. Good visualization on efficiency analysis.

Cons:
1. Please specify the values of λs in the equation.(5).
2. In Table.3, it would be necessary to see the effectiveness of the boundary supervision (GIou Loss of boundary).
3. In section 4.5, the authors mention,' While we found that too extreme augmentation may harm the performance of the classification head of the proposed architecture, it can benefit the contour proposal task.' The authors may need to provide more evidence or details in this paper. For example, which augmentations may harm the classification head but benefit the contour task?

---

### Official Review · Reviewer_JTxG · 2023-01-13
**A good paper**

**Rating:** 8
**Confidence:** 5

**Review:**

## Summary

This paper proposed an uncertainty-aware contour proposal network for multi-modality cell segmentation. The paper mainly based on the former Contour Proposal Network. To handle the issue in this competition, the paper proposed several improvements, including boundary supervision, localization uncertainty estimation,  uncertainty-aware NMS, and semi-supervised strategy Listen2Student. The experiments showed the outstanding performance of the framework.

## Pros:

+ The paper is well structed.
+ The proposed method is novel.

## Cons:

+ It is suggested to add a ablation study of localization uncertainty estimation to address the difference between UACPN and CPN.

---

### Decision · Program_Chairs · 2023-01-19

Accept